# Recent Developments in the Understanding of Immunity, Pathogenesis and Management of COVID-19

**DOI:** 10.3390/ijms23169297

**Published:** 2022-08-18

**Authors:** Aram Yegiazaryan, Arbi Abnousian, Logan J. Alexander, Ali Badaoui, Brandon Flaig, Nisar Sheren, Armin Aghazarian, Dijla Alsaigh, Arman Amin, Akaash Mundra, Anthony Nazaryan, Frederick T. Guilford, Vishwanath Venketaraman

**Affiliations:** 1College of Osteopathic Medicine of the Pacific, Western University of Health Sciences, Pomona, CA 91766, USA; 2Your Energy System, LLC 555 Bryant St. #305, Palo Alto, CA 94301, USA

**Keywords:** glutathione, coronavirus, SARS-CoV-2, COVID-19, vaccines, reactive oxygen species

## Abstract

Coronaviruses represent a diverse family of enveloped positive-sense single stranded RNA viruses. COVID-19, caused by Severe Acute Respiratory Syndrome Coronavirus-2, is a highly contagious respiratory disease transmissible mainly via close contact and respiratory droplets which can result in severe, life-threatening respiratory pathologies. It is understood that glutathione, a naturally occurring antioxidant known for its role in immune response and cellular detoxification, is the target of various proinflammatory cytokines and transcription factors resulting in the infection, replication, and production of reactive oxygen species. This leads to more severe symptoms of COVID-19 and increased susceptibility to other illnesses such as tuberculosis. The emergence of vaccines against COVID-19, usage of monoclonal antibodies as treatments for infection, and implementation of pharmaceutical drugs have been effective methods for preventing and treating symptoms. However, with the mutating nature of the virus, other treatment modalities have been in research. With its role in antiviral defense and immune response, glutathione has been heavily explored in regard to COVID-19. Glutathione has demonstrated protective effects on inflammation and downregulation of reactive oxygen species, thereby resulting in less severe symptoms of COVID-19 infection and warranting the discussion of glutathione as a treatment mechanism.

## 1. Introduction

In December of 2019, a rise in pneumonia cases of unknown origin were identified and linked to a wholesale market for seafood and wet animals in Wuhan, Hubei Province, China. Through the isolation and sequencing of human airway epithelial cells, the Chinese Center for Disease Control and Prevention (Chinese CDC) discovered a previously unknown betacoronavirus, coined 2019 novel CoronaVirus (2019-nCoV) [1]. The 2019-nCoV pathogen, now referred to as Severe Acute Respiratory Syndrome Coronavirus-2 (SARS-CoV-2), is responsible for causing the coronavirus disease infamously known around the world as COVID-19 [2]. Declared a global pandemic by the World Health Organization (WHO), SARS-CoV-2 is a highly contagious respiratory disease transmissible mainly via close contact and respiratory droplets, and less commonly through airborne, bloodborne, and animal-to-human modes of transmission. Clinical features vary from asymptomatic to flu-like symptoms (fever and cough) to severe pneumonia resulting in acute respiratory distress, multiorgan failure, and potentially death [3]. As of June 2022, the WHO reports over 528 million confirmed cases of SARS-CoV-2 and over 6 million SARS-CoV-2 related deaths globally [4]. In addition, the Centers for Disease Control and Prevention (CDC) reports a total of over 84 million confirmed cases of SARS-CoV-2 and over 1 million SARS-CoV-2 related deaths in the United States [5]. The information reviewed in this publication is intended to highlight a support for a host pathway, exploited by viruses for their replication. Liposomal glutathione offers a host directed adjunct for therapy that can be used against respiratory viruses.

## 2. Results

### 2.1. SARS-CoV-2

Coronaviruses (CoVs) are a diverse family of enveloped positive-sense single stranded RNA viruses and consist of four genera: alpha, beta, gamma, and delta [2]. Alpha and beta CoVs exclusively infect mammalian species, whereas gamma and delta CoVs are additionally capable of infecting avian species. Currently, there are seven CoV species that are known to cause human disease, referred to as human coronaviruses (HCoVs). Four of these HCoVs—HCoV-229E, HCoV-OC43, HCoV-NL63, and HCoV-HKU1—result in seasonal and mild respiratory tract infections similar to symptoms of the common cold. On the other hand, three HCoVs, i.e., Severe Acute Respiratory Syndrome Coronavirus (SARS-CoV) and Middle East Respiratory Syndrome Coronavirus (MERS-CoV), and most recently SARS-CoV-2, can result in severe, life-threatening respiratory pathologies via infection of bronchial epithelial cells, pneumocytes and upper respiratory tract cells [1,6].

The large genome of CoVs is packed within a helical capsid formed from the nucleocapsid protein (N) and surrounded further by an envelope. The envelope is further associated with three additional structural proteins. The membrane (M) protein and the envelope (E) protein are responsible for virus assembly. The club-shaped spike (S) protein projections extend from the CoVs spherical surface and are responsible for entrance of the virus into host cells [7]. The S protein consists of three segments: an ectodomain, a single-pass transmembrane anchor, and a short intracellular tail. The ectodomain further consists of a subunit responsible for receptor-binding (S1) and a subunit responsible for membrane-fusion (S2). Following attachment of the receptor binding domain (RBD) of the S1 subunit to its angiotensin-converting enzyme 2 (ACE2) receptor on the host cell surface, the S2 subunit fuses the host and viral membranes resulting in entrance of the viral genome into human cells [2,7]. Upon entrance to the cytoplasm of the host cell, the products of replication, transcription, and translation of the viral structural proteins are assembled and released via exocytosis, allowing for the virus to continue to spread within an infected organism and to other humans via horizontal transmission [2].

### 2.2. Evolving into Multiple Variants

Due to its ability to adapt to environments through mutations and recombination, and therefore altering host range and tissue tropism, SARS-CoV-2 has evolved into numerous variants referred to as variants of concern (VOCs). Evolution of CoVs is made possible due to their large genome, high mutation rate, and high recombination frequency. Mutations that result in greater fitness are selected for, resulting in VOCs. VOCs are recognized by their increased transmissibility, immuno-escape from neutralizing antibodies or T-cell immunity, and pathogenicity [7,8,9]. According to the WHO, to be classified as a SARS-CoV-2 variant, the virus of concern must demonstrate one or more of the following: increase in transmissibility or detrimental change in SARS-CoV-2 epidemiology, increase in virulence or change in clinical disease presentation, and/or decrease in effectiveness of public health and social measures, available diagnostics, vaccines, or therapeutics. The WHO recognizes the following five variants of SARS-CoV-2: alpha (PANGO lineage B.1.1.7), beta (PANGO lineage B1.1.351), gamma (PANGO lineage P.1), delta (PANGO lineage B.1.617.2), and most recently omicron (PANGO lineage B.1.1.529) [10].

The alpha variant, which emerged out of the United Kingdom in September 2020, is understood to have functional mutations to the S protein resulting in increased transmission and increased resistance to antibody-mediated neutralization [9,11]. First appearing in South Africa in October of 2020 is the beta variant, which has functional mutations to the S protein, similarly resulting in increased transmission [9,12]. The gamma variant emerged out of Brazil in July of 2020 and is believed to have mutations in the RBD of the S protein resulting in increased transmission. However, less is understood about this variant [9,13]. The delta variant, emerging out of India in October of 2020, contains mutations to the S protein resulting in increased electrostatic interactions between the RBD and ACE2 receptor (RBD-ACE2), resulting in an increase in RBD-ACE2 stability and therefore increased viral infectivity and virus replication. In addition, the delta variant is noted to have increased resistance to monoclonal antibodies, convalescent sera, and vaccinated sera, all of which have been vital treatments for CoVs infections [9]. The Omicron variant, emerging out of multiple countries in November of 2021, is characterized by mutations to the S protein, similar to the Delta variant, leading to increased binding affinity between the RBD and ACE2 receptor and ultimately increased spread of the virus [8,10]. The omicron variant has undergone further changes, leading to descendent lineages with different genetic makeup. This has led to reduction in neutralizing titers post immunization or prior infection, with currently available vaccines. However, data shows significant reduction in hospitalization rate with Omicron infection compared to the Delta variant [10].

### 2.3. Increased Susceptibility to COVID-19 among Risk Groups

Although SARS-CoV-2 virus has been infectious to all age groups, regardless of their health status, age and pre-morbidities have been shown to increase the risk of adverse consequences when infected [14]. Comorbidities, such as diabetes, chronic obstructive pulmonary disease (COPD), cardiovascular diseases (CVD), hypertension, malignancies, human immunodeficiency virus (HIV), and obesity, among others, have also been correlated with increased risk of severe symptom presentation when infected with SARS-CoV-2 [14,15]. Individuals with certain comorbidities highly express surface ACE2 receptors on cells [15], which bind the S1 protein found on SARS-CoV-2 virus [14]. Once bound, the S2 protein on SARS-CoV-2 facilitates the fusion of viral membrane with host cell membrane [14]. The high release of proprotein convertase in individuals with comorbidities facilitates the endocytosis of the virus into the host cells [15]. The enhanced facilitation of viral entry into host cells in individuals with comorbidities leads to significantly higher risk of morbidity and mortality [15]. People suffering from diabetes mellitus (DM) and/or obesity have impaired innate and adaptive immune systems which produce more pro-inflammatory cytokines and fewer anti-inflammatory cytokines than healthy individuals [16]. This suboptimal level of immunity can be further exacerbated when infected with SARS-CoV-2, leading to higher mortality and morbidity among these individuals [16].

### 2.4. Pathogenesis of the Disease Induced by Cytokine Storm, Increased IL-6, Decreased Glutathione

It appears that all viruses deplete reduced glutathione (GSH), an intracellular antioxidant, as part of their strategy for replication [17]. In a study designed to analyze the different pathways exploited by viruses for replication, the influenza viruses increased the production of reactive oxygen species (ROS) and decreased the production of GSH. This led to a state of oxidative stress [18,19,20]. In response to an infection, the body releases Interleukin-6 (IL-6) and other proinflammatory cytokines. As shown in Figure 1, IL-6 blocks the enzymes Iodothyronine deiodinases type I (D_1_) and II (D_2_) responsible for conversion of prohormone thyroxine T4 to active T3 [21]. In astrocytes, thyroid hormones upregulate glutamate cysteine ligase (GCL), which enhances GSH synthesis [22]. Therefore, blocking thyroid synthesis due to a viral infection can suppress GSH synthesis and lead to increased oxidative stress in cells.

Nicotinamide adenine dinucleotide phosphate (NADPH) oxidase (NOX) family consists of seven members: NOX1 to NOX5 and the two dual oxidases, Duox1 and Duox2, which are expressed in most cell types [23]. The NOX family mediates ROS production in cells which plays an important role in the survival of these cells. During a viral infection, inflammatory cells release NOX2 which enhances the pathogenicity of viruses, similar to influenza A viruses [24]. It has been shown that inhibition of NOX2 oxidase activity lessens influenza A virus-induced lung inflammation in mice [25]. GSH acts as the main intracellular antioxidant against ROS through oxidation of the thiol group of its cysteine to a disulfide bond (GSSG), hence reducing the oxidized species and maintaining the redox state of the cell. GSSG is then converted back to GSH through the action of glutathione reductase. GSH also neutralizes the potentially harmful metals and xenobiotics in the body [26]. In addition, GSH can act as a signaling molecule to the innate immune system [26,27,28]. The decrease in GSH levels in infected individuals allowed viral glycoprotein haemagglutinin folding and maturation and hence viral replication [29].

Elevated serum concentrations of IL-6 and TGF-β were noted in patients with severe SARS-CoV-2 symptoms compared to those with less severe symptoms [30]. It has been shown that IL-6 induced a dose-dependent decrease in intracellular GSH levels in a number of human cell lines, including lung cells [21]. Additionally, an animal study has shown that IL-6 can down regulate the expression of Nrf2 target genes resulting in decreased expression of the enzyme GCL, which is needed to form GSH in cells, leading to a decreased availability of GSH [31]. Research showed that transforming growth factor β (TGF-β) is present in SARS-CoV-2 patients and can be classified as a strong predictor of SARS-CoV-2 disease severity [32]. Notably, high levels of monocyte chemoattractant protein-1 (MCP-1) and TGF-β1 were identified in an autopsy for SARS-CoV-2 infected lung cells [33]. It has been shown that TGF-β suppresses GCL gene expression and induces oxidative stress in a lung fibrosis model [34]. GCL controls GSH formation intracellularly [35]. The elevation of TGF-β adds additional suppression of GSH formation during SARS-CoV-2 infection. High levels of TGF-β1 may trigger fibrotic changes and may account for the typical early CT scan features of SARS-CoV-2 pneumonia such as the “ground-glass opacity” appearance [36,37]. Fibrosis is a major complication related to SARS-CoV-2 infection [37,38] and is related to TGF-β elevation [34]. The high levels of MCP-1 explain the findings of a mononuclear inflammatory infiltrate with lymphocytes and CD68+ macrophages in the lungs of SARS-CoV-2 patients [38,39,40,41].

Both respiratory syncytial virus (RSV) and SARS-CoV-2 infections deplete GSH and decrease the formation of the enzymes needed to form GSH in the cells they invade [17], which occurs in immune cells as well as other cells such as lung Type II pneumocytes. Type II pneumocytes are responsible for the production of GSH, which is found at a very high level in the epithelial lining fluid (ELF) of the lung [42]. When SARS-CoV-2 reaches the lower respiratory tract, it will bind to ACE2 on Type II pneumocytes, leading to viral infection and decreased function. When the Type II pneumocyte function is compromised by viral infection, the level of GSH in the ELF can become diminished, which is associated with compromise of lung function. This can lead to acute respiratory distress syndrome (ARDS) [43], which is associated with SARS-CoV-2 infection [44]. SARS-CoV-2 ARDS appears to have worse outcomes [44]. Type I pneumocytes are supported by Type II pneumocytes, and damage leads to decreased function of the Type I pneumocyte pool. Damage to Type I pneumocytes leads to impaired gas exchange (1). This compromise of gas exchange with a buildup of carbon dioxide tension (Pa_CO2_) blunts the brain’s response to hypoxia and leads to the presentation of individuals with “silent hypoxia.” Thus, patients with SARS-CoV-2 are described as exhibiting oxygen levels incompatible with life without dyspnea [45].

### 2.5. Increased Susceptibility to TB Due to Decrease in Glutathione

Research has shown that the survival of intracellular *Mycobacterium tuberculosis* (Mtb) is enhanced due to the increased uncleared free radicals and inflammatory cytokines as a result of GSH depletion [46]. Therefore, immunocompromised individuals, such as those with Human immunodeficiency virus (HIV) and Type 2 Diabetes Mellitus (T2DM), are at higher risk of contracting Mtb infection [46]. HIV infection is associated with an accumulation of ROS [47] mediated by the envelope protein gp120 [48] and Tat proteins [49]. Furthermore, NOX2 and NOX4-induced ROS overproduction has been reported in HIV gp120 treated astrocytes [50]. An excess of ROS persists in HIV+ individuals even after successful HAART therapy and results in a depletion of GSH [35]. It appears that the decrease in GSH is a result of the production of cytokines including IL-1, TNF-α and IL-17, and TGF-β [35]. TGF-β interferes with the biosynthesis of GSH [51]. Elevation of both IL-6 and TGF-β has also been shown to accompany the loss of GSH in HIV+ individuals and to be decreased by the administration of liposomal GSH [52,53].

According to research, immunological abnormality plays a significant role in the increased susceptibility to Mtb infection in T2DM individuals [54]. Through innate immunity, the elevated release of cytokines in diabetic individuals leads to a decrease in GSH production, and hence an increase in oxidative stress in cells [35]. In addition, compared to healthy individuals, people with T2DM have lower levels of GCL, the rate-limiting enzyme in GSH synthesis [46]. This leads to a significantly lower plasma GSH:GSSG ratio in diabetic patients when compared to controls [55]. Consequently, ROS accumulate in cells, and have been associated with further complications of T2DM [55]. Through adaptive immunity, research revealed that a decrease in Th1:Th2 cytokines can increase the susceptibility to Mtb infection in T2DM individuals [54]. However, maintenance of normal GSH levels promotes Th-1 differentiation via IL-12 and IL-27 cytokines that were otherwise downregulated in immunocompromised patients [46].

### 2.6. Modulation of Expression of Antioxidant Genes

Nuclear factor erythroid 2p45-related factor 2 (Nrf2) is a transcription factor, which has evolved as an oxidant-sensitive molecule that is activated and will transcriptionally stimulate a series of genes responsible for cytoprotection and detoxification [56]. Nrf2 is one of the best-characterized antioxidative transcription factors with an oxidant/electrophile-sensor function [57]. Under normal conditions, it forms a complex with Kelch-like ECH-associated protein 1 (KEAP1), a well-known negative regulator of Nrf2 [58]. Since KEAP1 serves as an adaptor protein for cullin-3-based E3 ubiquitin ligase, this dimeric Nrf2/KEAP1 complex subjects Nrf2 to constant ubiquitination and proteasomal degradation [59]. In regard to their oxidant-sensing mechanisms, redox-sensitive twenty-five cysteine residues of KEAP1 were shown to play a key role in the regulation of the E3 ubiquitin ligase activity [60]. Essentially, these cysteine residues are very susceptible to conjugation of a variety of ROS-inducing agents. Once conjugated, the KEAP1-mediated ubiquitination of Nrf2 is severely diminished [61]. This leads to liberation of Nrf2 from the KEAP1-mediated restraint. Once stabilized, Nrf2 is able to get inside the nucleus and form complexes with one of small musculoaponeurotic fibrosarcoma proteins and other coactivators. As shown in Figure 2, binding of this trimeric complex to the antioxidant response elements (AREs) in the promoter regions facilitates the transcription of a series of cytoprotective and detoxifying genes, such as heme oxygenase-1 (HO-1), NAD(P)H quinone oxidoreductase-1 (NQO-1), GCL catalytic subunit (GCLC) and GCL regulatory subunit (GCLM), glutathione S-transferase (GST), uridine diphosphate glucuronosyltransferase (UDPGT), superoxide dismutase isoform 1 (SOD1), catalase (CAT), glucose 6 phosphate dehydrogenase (G6PD), and glutathione peroxidase-1 (GPx) [62,63,64,65]. Importantly, GCLC and GCLM are critical components of the production of GSH [35].

Viruses possess a variety of adaptive mechanisms to deplete GSH in host cells. The HIV virus can decrease the expression of GCLC and GCLM in HIV+ macrophages to about half and GSH is known to be deficient in individuals with HIV [35,66]. Respiratory Syncytial Virus (RSV) was shown to use NOX2 as an essential regulator of RSV-induced NF-κB activation. RSV virus infection led to continuous activation of NF-κB, which likely caused excessive NF-κB-mediated inflammatory gene expression [67]. A later study found that RSV infection down-regulates Nrf2 expression in airway epithelial cells and that a decrease in the expression of airway antioxidant enzymes led to additional oxidative stress [68]. Nrf2 mRNA levels were decreased following RSV infection and the nuclear localization of the protein was decreased in infected cells compared to uninfected ones, resulting in increased oxidation stress and a significant decrease in the GSH:GSSG ratio [69].

In COVID-19 infection, depletion of GSH begins with the binding of SARS-CoV-2 S protein to ACE2, which results in inhibition and decrease of ACE2 expression in infected cells and leads to toxic overaccumulation of ANGII [70]. The increased ANGII, through binding to AT1R, activates NADPH oxidases (NOX) that transfer an electron from NADPH to O_2_, generating several radical species which are scavenged by GSH and deplete GSH [70]. As GSH is depleted, ROS-mediated oxidation increases. Although SARS-CoV-2 proteins are synthesized in the cytosol, some viral proteins are also detectable in the nucleus, including Nsp1, Nsp5, Nsp9, Nsp13, Nsp14, and Nsp16 [71]. This becomes important as recent research has shown that the S protein also localizes in the nucleus and inhibits DNA damage repair by impeding key DNA repair protein BRCA1, and recruitment of 53BP1 to the damage site [72].

BRCA1 is known as a tumor suppressor as mutations in this gene confer an increased risk for breast, ovarian, and prostate cancers [73]. Lesser well known is BRCA1′s function in stimulating the antioxidant response element (ARE), driving transcriptional activity and its enhancement of the antioxidant response transcription factor Nrf2 [73]. Thus, protecting against oxidative stress is important in BRCA1′s role as a caretaker gene [73]. BRCA1 increased the expression of some of these Nrf2-regulated genes (NQO1, MGST1/2, Gsta2, G6PD, and ME2), and BRCA1 also induced (and BRCA1 mutation inhibited) expression of glutathione peroxidase (GPX3) [73]. In addition to endogenous reactive oxygen species, which contribute to carcinogenesis, many DNA-damaging agents and xenobiotics cause oxidative stress, resulting in DNA damage, protein oxidation, and lipid peroxidation [74]. Some of these lesions are detoxified by BRCA1-regulated genes (e.g., GSTs, GPXs, oxidoreductases, and paraoxonases) [73]. BRCA1 stimulates GSH production under oxidizing conditions, possibly by increasing the levels of glucose-6-phosphate dehydrogenase (G6PD). G6PD can then stimulate NADPH production, which plays a role in the regeneration of GSH [73].

These findings suggest the S protein may decrease both the intracellular and intranuclear antioxidant protection of GSH. If the S protein of SARS-CoV-2 localizes to the nucleus and decreases BRCA1 recruitment and thus the benefit of Nrf2 function, evidence of oxidation of nuclear material might be evident. A decrease in available BRCA1 may contribute to the finding of increased oxidized guanine species that have been identified at higher levels in the serum of non-surviving individuals than in those surviving SARS-CoV-2 [75]. Additionally, it was shown that IL-6 induced a dose-dependent decrease in intracellular GSH levels in a number of human cell lines, including lung cells [21]. IL-6 was shown to increase the levels of superoxide radicals, leading to the decrease of intracellular GSH [21]. Increased expression of IL-6 is seen in severe SARS-CoV-2 patients, and its levels are even used to gauge the severity of the disease [76]. An animal study showed that IL-6 can down regulate the expression of Nrf2 target genes resulting in decreased expression of the enzyme glutamyl cysteine ligase (GCL), leading to decreased availability of GSH [31]. These findings highlight the importance of IL-6 and the S protein in maintaining a pro-inflammatory state and reducing the antioxidant protection mechanisms needed to deal with ROS.

### 2.7. COVID-19 Can Increase the Risk for Development of TB

As previously mentioned, the S protein of SARS-CoV-2 plays a significant role in the viral pathogenicity and its downstream consequences. When BRCA1 is inhibited, ARE production is downregulated, resulting in decreases in antioxidant molecules such as GSH [77]. Due to the excessive oxidative stress caused by SARS-CoV-2 infection, cells have a diminished capacity to undergo de novo GSH synthesis. GSH depletion has been associated with increased risk of opportunistic infections such as Mtb [77]. Large increases in ROS, in addition to GSH depletion, place individuals at increased risk of Mtb. The ability to ward off active Mtb is dependent on CD4+ T-cells. These cells play an integral role in adaptive immunity as well as providing “memory”, to fight off repeat infections more effectively. Opportunistic infections like HIV and Mtb infections have been shown to rely heavily on CD4+ T-cells, which serve as predictors for risk of infections [78]. Both Mtb and SARS-CoV-2 infections invade lung pneumocytes. Patients infected with SARS-CoV-2 have an increased susceptibility to Mtb. SARS-CoV-2 inhibits several mediators which can help fight off Mtb infection. Under normal circumstances, Mtb enters respiratory airways as either droplets or aerosols and ultimately ends up in the alveoli. At this point the bacteria is phagocytosed by alveolar macrophages (AMs). The pathogenicity of Mtb has been well described in the literature and includes the interruption of key signaling pathways including cyclic GMP-AMP synthase (cGAS), stimulator of interferon genes (STING), which involve interferons type I and II. Type II interferon, also called interferon-gamma (IFN-ɣ), is the major cytokine involved in protection against Mtb infections [79].

Management of Mtb infection involves both innate and adaptive immune cells. The first line of defense includes phagocytic cells which are found in the lung epithelium. Integral adaptive immunity responses include CD4+ T-cells that produce IFN-ɣ, which serves to activate AMs [80]. These AMs within the lung epithelium serve as an important entry point for SARS-CoV-2 viruses as they possess the ACE2 receptors. In the presence of SARS-CoV-2 infection, these AMs become damaged and are less capable of producing normal amounts of IFN-ɣ [81]. A specific type of SARS-CoV-2 infection, called ORF9b, involves a unique accessory protein (ORF9b) that antagonizes the cGAS-STING pathway and leads to a decreased ability to combat many inflammatory processes, including but not limited to infections [82]. As shown in Figure 3, SARS-CoV-2 ORF9b inhibits phosphorylation of TANK binding kinase 1 (TBK1, a serine/threonine protein kinase) leading to the inhibition of IRF3 and its nuclear translocation, and hence type I interferon transcription. The antagonism of the cGAS-STING pathway ultimately leads to a down regulation of de-novo interferon I synthesis, as well as IFN-ɣ which is produced by activated lymphocytes, including AMs [83]. With these decreased levels of both IFN-ɣ and cGAS-STING pathway activity, Mtb is better able to proliferate in the presence of SARS-CoV-2 infection. Additionally, Mtb is able to exit phagosomes via ESX-1, a type VII secretion system utilized by the bacteria [84]. Once in the cytoplasm, the bacterial DNA can be recognized by the cGAS-STING pathway and upregulate type I interferons and IFN-ɣ. One study in mice found that cGAS-STING pathway is greatly increased during Mtb infections in vitro, which is critical for the pulmonary expression of IFN-β (a type I interferon) [85].

Overall, there are several key factors that are negatively affected by SARS-CoV-2 infection, including GSH synthesis, IFN-ɣ and cGAS-STING pathways. The interference with these pathways by SARS-CoV-2 infection places patients at a greater risk for the development of Mtb. It is important to note the concurrent infections of SARS-CoV-2 and Mtb, as they are both respiratory illnesses, can rapidly spread and have large socioeconomic impacts on a global scale. The mortality rate of Mtb and SARS-CoV-2 coinfections is 12.3%, which is much higher than SARS-CoV-2 infection alone [86]. Mtb is the second leading cause of deaths caused by respiratory illness, after SARS-CoV-2 [87]. With a large number of Mtb infections going unnoticed, it would be beneficial to improve screening and testing for latent Mtb to avoid more severe cases of people developing concurrent Mtb and SARS-CoV-2 infections. As previously mentioned, SARS-CoV-2 primarily infects the respiratory epithelium, subsequently invading and activating alveolar macrophages (AMs) [88]. AM activation occurs via signals such as IFN-ɣ and causes the release of a host of pro-inflammatory cytokines such as IL-6, which has been shown to be elevated in SARS-CoV-2 infected individuals.

### 2.8. Thrombus Formation during SARS-CoV-2 Infection

One potential major complication of more severe stages of SARS-CoV-2 infection is the increased risk of thrombus formation. GSH serves an integral role in protecting against ROS, helping to lower ACE activity, and has demonstrated the ability to downregulate the production of Nuclear Factor Kappa B (NF-kB) [71]. NF-kB serves as an important mediator of inflammation and coagulation. Vascular endothelium, monocytes, and even neutrophils all respond to NF-kB in a pro-coagulatory manner and contribute to thrombus formation [89]. NF-kB signaling leads to numerous physiological changes, including vascular endothelial changes promoting adhesion molecules that allow leukocyte migration into tissues. Additionally, neutrophils can be signaled to release their DNA from the cell and become neutrophil extracellular traps (NETs). It is thought that NETs’ main function is to trap and kill microbes, but this has also been associated with clot formation [89]. These NETs interact with the complement system and interestingly also provide a framework for thrombus formation to begin [90]. In the pulmonary vasculature, the hyperinflammatory response seen with SARS-CoV-2 infection results in the accumulation and localization of NETs along with platelets, which contribute to thrombus formation. One suggested treatment to reduce these NET/platelet complexes and even reduce the activation of endothelial cells is by the administration of N-acetyl cysteine (NAC), a precursor molecule for GSH [89].

Matrix metalloproteinases (MMPs) are important enzymes involved in processes including degradation of the extracellular matrix (ECM) and can be activated by NF-kB [91,92]. These MMPs are widespread throughout the body and are important in epithelial tissues, such as the gastrointestinal tract, respiratory system, and vascular endothelium. As they relate to vascular endothelium, MMPs degrade the ECM to allow for vessel remodeling under both normal and pathological conditions. With MMPs activated, epithelial cells become more permeable, allowing for healthy tissue turnover, but also lends the opportunity for pathogenesis, including cardiovascular diseases and infiltration by host defenses such as NET/platelet complexes. In one study looking at patients with systemic lupus erythematosus (SLE), MMP-2 inhibition resulted in lower levels of NETs and improved endothelial function [93]. We can see that when GSH levels are within normal ranges, less NF-kB will be produced, ultimately leading to lower levels of MMPs and NETs, which can prevent thrombus formation.

Furthermore, during high oxidative stress situations, such as during SARS-CoV-2 infection, RBCs can adhere to the endothelial lining of blood vessels and contribute to the clot formation [94]. When GSH levels are low, or there is a low GSH:GSSG ratio, platelet aggregation is potentiated [95]. There is evidence that additional factors involved in the clotting cascade become elevated during SARS-CoV-2 infection, such as von Willebrand factor, Factor III (tissue factor), and free thrombin [96]. Romagnoli et al. (2012) found that correction of the GSH:GSSG ratio led to decreased production and activity of MMP-2 in intestinal subepithelial myofibrobalsts (ISEMFs) [97]. This suggests that restoration of normal levels of GSH (or GSH:GSSG ratio) can serve as protection against exaggerated increases in MMP-2 levels and other elements that relate to the clotting cascade.

Lower GSH levels are associated with elevation in many clotting elements and pro-coagulatory components, such as NETs, MMPs, NF-kB, and vessel endothelial changes, placing patients at a greater risk for microvascular thrombosis and platelet aggregation. Thus, it is important to maintain adequate levels of GSH as it can protect against thrombosis in more severe forms of SARS-CoV-2 infections.

### 2.9. Vaccines, Vaccine Evasion, and Protection against COVID-19

Development of the major vaccines used for SARS-CoV-2 were based on the experience with severe acute respiratory syndrome coronavirus (SARS-CoV) and Middle East respiratory syndrome coronavirus (MERS-CoV). Currently, there are multiple vaccines that have emergency-use approval in a large number of countries, which differ in their type and technique used to elicit an immune response. There are mRNA vaccines, such as BNT162b2 (Pfizer) and mRNA-1273 (Moderna), and adenoviral-vectored vaccines, including Ad26.COV2.S (Janssen) and ChAdOx1nCoV-19 (University of Oxford/AstraZeneca), all of which focus on providing antibodies against the S protein of SARS-CoV-2 [98,99,100]. As previously mentioned, the S protein allows the virus to bind to the ACE2 receptor and enter its host cell, making the S protein an important target in vaccine development [101]. Once antibodies have been created against the S protein, the body can produce a robust immune response to fight off future infection.

The immune response to SARS-CoV-2 in humans involves both humoral and cell-mediated immunity [102,103]. Antibodies directed at the S protein increase the chances of survival, and neutralizing antibodies (NAbs) targeted at the S protein are present in most individuals [104]. Studies have shown that the amount of NAbs directed at the S protein is a strong indicator for survival, and it is imperative for vaccines to elicit NAb responses [105,106]. Indeed, both mRNA vaccines, the BNT162b2 and mRNA-1273, have been shown to increase levels of RBG-binding IgG antibodies and neutralizing antibody titers when compared to those without the vaccine [107,108]. These antibodies play an important role in neutralizing and preventing viral infection and have been correlated with increased survival [109]. In addition, BNT162b2 can stimulate the production of antiviral cytokines, such as IFN-γ, which has been shown to inhibit the replication of SARS-CoV-2 [110,111]. Both mRNA vaccines elicit a Th1 immune response, with levels of antigen specific CD8+ memory cells and CD4+ T-cells greatly increased [112]. Clinical trials with these mRNA vaccines showed two doses of BNT162b2 and mRNA-1273 were 95% and 94.1% effective at preventing the original SARS-CoV-2 virus, respectively [113,114].

The Ad26.COV2.S and ChAdOx1nCoV-19 vaccines are adenoviral-vectored that target the full-length S protein of SARS-CoV-2. Overall, they provide a very similar immune response as the mRNA vaccines, with both adenovirus vaccines increasing levels of spike antibodies and RBG specific IgG, IgA, and IgM antibodies [115,116]. Likewise, they also induce a largely Th1 cellular immune response, increasing the levels of CD8+ and CD4+ T-cells to fight off current infection and confer immune memory for future infection [115,116]. The Ad26.COV2.S vaccine has a single dose and showed a 69.7% efficacy against SARS-CoV-2, while ChAdOx1nCoV-19 is two doses giving a 74% efficacy against the virus [117,118].

Even with the development of multiple vaccines, research has shown their efficacy decreases five months after initial vaccination [119]. This has led to the creation of additional booster shots that are given at this time to provide more immune protection. They have been shown to be effective at increasing levels of antibodies against SARS-CoV-2, with the BNT162b2 and mRNA-1272 booster shots giving higher protection than their original vaccines [119]. Furthermore, despite these recent advancements in vaccine technology, there have been mutations in the SARS-CoV-2 virus that have helped it evade vaccines. These recombination events surrounding the emergence of vaccine evading variants suggests that these changes could also lead to the virus evading pharmacologic therapies.

Viral escape mutations on the S protein receptor-binding domain (RBD) provide us with insights into infectivity and antibody resistance caused by new variants of SARS-CoV-2 [101]. RBD site facilitates binding between the S protein and the host ACE2. One example of a variant with mutations on the S protein is the highly transmissible B.1.1.529 Omicron (Omicron) variant of SARS-CoV-2. The Omicron variant has been shown to exhibit as many as 15 mutations at the RBD site and 32 on the S protein, allowing for stronger binding to the ACE2 receptor and the ability to evade current vaccines [101].

A recent study tested the ability of monoclonal antibodies (mAbs) to neutralize an infection Omicron isolate. Several mAbs lost their neutralizing activity, and several others had reduced capacity to neutralize the isolate [120]. This study shows that some antibodies in clinical use could lose efficacy against emerging variants. This highlights the importance of effective responses against a wide range of epitopes on the S protein and surveillance of vaccine effectiveness in response to emerging variants.

### 2.10. Vaccine Hesitancy

In addition to antibody resistance and escape mutations that emerging variants thrive upon, vaccine hesitancy is also another burden towards achieving herd immunity, in which a large enough proportion of a community, area, or location are immune to a specific cause of a disease, such as a virus or bacteria. According to The SAGE Working Group, vaccine hesitancy is the delay in acceptance or refusal of vaccination regardless of the availability or accessibility of vaccination services [121]. Hesitancy can manifest itself in a wide spectrum of actions, including refusal of some vaccines but agreeing to others, delaying vaccination, or accepting vaccination but not sure in doing so. Thus, the term hesitancy is considered to be a continuum between those individuals that accept all vaccines without doubts, to individuals who completely refuse vaccines with no doubts, with hesitant individuals falling in between these two extremes [122].

Vaccine hesitancy can increase the risk of vaccine-preventable disease outbreaks and epidemics and also delay achieving herd immunity even when there are no accessibility or availability issues [122]. Even though vaccination is considered as one of the most successful public health measures, it is still perceived as unsafe and unnecessary by some individuals. There are socio-cultural aspects to vaccine hesitancy, which includes conversations regarding controversies and vaccination scares, media, and other aspects of daily living that encompass some individuals.

In today’s world, technology, particularly media, plays a powerful role in spreading misinformation and causing fear. One example of an extensive and large-scale vaccination scare that technology allowed to become widespread was the correlation between vaccines and autism [122]. The misconception surrounding this correlation led to difficulties for the scientific community to educate the public effectively and adequately about precautionary health measures. These types of doubts towards the scientific community regarding vaccination persist throughout the world and pose challenges towards reaching herd immunity early on and preventing epidemics as efficiently as they are possible.

Furthermore, developing countries possess additional challenges in addition to vaccine hesitancy. Many developing countries rely on vaccines developed in other nations due to economic burdens and lack of technological advancements. Thus, one of the best options is buying from other developed nations, but most global vaccine availability is also taken up by wealthier nations in the beginning. Thus, this pushes developing nations to the back of the list and prolongs wait time for accessing vaccines [123]. Some of the vaccines also require extra safety measures for viability. Some require cold temperatures for storage and transfer, and thus require more advanced technology and equipment for viable administration and vaccine services [124]. It is important to implement the lessons learned here in the future, to prevent epidemics and expedite global cooperation and information sharing.

### 2.11. Newly Approved Pills for SARS-CoV-3 and Their Adverse Effects

Since the onset of the SARS-CoV-2 pandemic, there has been extensive research and discoveries made in the search to find therapies for the prevention of, and progression to, severe SARS-CoV-2 infections. Different classes of drugs have implemented potential targets of the virus, and of these, nirmatrelvir, remdesivir, molnupiravir, baricitinib, and sotrovimab have shown promising results. They have gained Food and Drug Administration (FDA) approval for use in SARS-CoV-2 affected patients through the Emergency Use Authorization (EUA) and/or eventually filing for a New Drug Application to be continuously used [125].

Pfizer has recently been granted an emergency use authorization (EUA) to make a novel oral antiviral drug candidate under the name of PAXLOVID, consisting of nirmatrelvir (PF-07321332) and ritonavir tablets, available for the treatment of SARS-CoV-2. It is used to treat mild-to-moderate SARS-CoV-2 infections in adults and children (12 or older and weighing at least 88 pounds) with a positive test for SARS-CoV-2 and who are at risk for progression to severe SARS-CoV-2 [126]. Nirmatrelvir primarily functions as a SARS-CoV-2 protease inhibitor that inhibits viral replication at the stage of proteolysis by targeting the SARS-CoV-2 3-chymotrypsin-like cysteine protease enzyme (Mpro) right before DNA replication [127,128]. It is designed to be administered orally at the first sign of exposure. Nirmatrelvir co-administered with a low dose of ritonavir has shown to slow the metabolism of the drug in the human body, allowing it to be more effective in combating the virus efficiently. Nirmatrelvir is mainly metabolized by CYP3A4, and therefore co-administration with a CYP3A4 inhibitor such as ritonavir enhances its pharmacokinetics [129].

Results from phase 2–3 trial in unvaccinated individuals given nirmatrelvir (300 mg) with ritonavir (100 mg) every 12 h for five days showed an 89.1% relative risk reduction in SARS-CoV-2 related hospitalization deaths from any cause, and an additional reduction in SARS-CoV-2 viral load at day 5 by a factor of 10 compared to placebo [130]. The most frequent adverse effects reported were dysgeusia, diarrhea, myalgia, and hypertension [131]. Treatment of symptomatic SARS-CoV-2 patients with nirmatrelvir plus ritonavir lowers the risk of progression to severe SARS-CoV-2 without any evident safety concerns and is a good candidate drug for first line treatment of SARS-CoV-2 in symptomatic individuals. Although PAXLOVID is currently authorized for emergency use, Pfizer announced on 30 June 2022, the submission of a New Drug Application to the U.S. Food and Drug Administration (FDA) for the approval of PAXLOVID for patients who are at high risk for progression to severe illness from SARS-CoV-2 after recording available safety data that are consistent in more than 3500 PAXLOVID-treated participants across the EPIC clinical development program [132].

Veklury (remdesivir) was approved by the FDA on 25 April 2022, for the treatment of SARS-CoV-2 in adults and pediatric patients over the age of 28 days who are either hospitalized with or have mild-to-moderate SARS-CoV-2 and are considered to be high risk for severe progression of SARS-CoV-2 [133]. Veklury is a prodrug of an analog to adenosine nucleoside triphosphate that targets the viral RNA dependent RNA polymerase [134,135]. In human airway epithelial cell assays, remdesivir has been shown to inhibit replication of coronaviruses. In mouse infection models, it was effective therapeutically against SARS-CoV-2 [136].

For adults and pediatrics weighing ≥40 kg, remdesivir should be initiated as early as possible following diagnosis of symptomatic SARS-CoV-2 and within seven days of symptom onset. Treatment should begin with 200 mg on Day 1 followed by a once-daily maintenance dose of 100 mg from day 2 onwards to be administered intravenously. Treatment duration varies based on SARS-CoV-2 severity status ranging from three to 10 days. The most common adverse reactions in ≥5% were nausea, headache, cough and increases in the liver enzymes, alanine transaminase (ALT) and aspartate transaminase (AST). The most serious side effect of this drug is renal impairment and is not recommended in persons with an eGFR < 30 mL/min [133,137].

For non-hospitalized patients with SARS-CoV-2, a three-day course of remdesivir was found to decrease the risk of SARS-CoV-2 related hospitalization and/or death by 87% [137]. The Adaptive SARS-CoV-2 Treatment Trial (ACTT) was implemented to study remdesivir for SARS-CoV-2 treatment. According to the ACTT, hospitalized patients who received a 10-day course of remdesivir were found to have a shorter median time to recovery (10 days) than those in a placebo group (15 days) as well as lowering the risk of progression to mechanical ventilation [138]. In the pathway towards finding ways to manage SARS-CoV-2, remdesivir adds another tool to help combat the progression of high-risk SARS-CoV-2.

Molnupiravir is an RdRp (RNA-Dependent RNA-Polymerase) inhibitor. In December 2021, the drug was granted FDA emergency use authorization for the treatment of mild to moderate SARS-CoV-2 patients with at least one risk factor for progression, and for whom other FDA approved treatments are not available. This authorization was granted based on the results of the MOVe-OUT study which showed that treatment with Molnupiravir, initiated within 5 days after the onset of symptoms, reduced the risk of hospitalization or death from SARS-CoV-2 [139]. Molnupiravir exerts its effect through its active form, EIDD-2061, being utilized by RdRp as a substrate in place of cytidine/uridine triphosphate, resulting in the generation of mutated RNA copies that leave the virus unable to replicate [140]. Molnupiravir is administered orally and well absorbed with minimal interference from food, making it useful in an outpatient setting. Results from Phase 1 studies, such as that of Painter et al. (2021), demonstrated that Molnupiravir is well tolerated, with mild adverse events mainly consisting of diarrhea and headache [141]. No accumulation was seen in multiple-ascending-doses and little of the drug or its metabolites were detected in urine. Hiremath 2021 further affirmed that Molnupiravir has a strong safety profile [142]. Although the mutagenic antiviral activity of Molnupiravir is speculated to be a risk for being incorporated into human DNA, several preclinical studies concluded that treatment with Molnupiravir cannot induce mutations in host cell DNA [136]. Of note, the FDA authorization letter stated that Molnupiravir should not be used in pregnant women due to the fetal toxicity reported in preclinical studies. The letter also reports that Molnupiravir may cause adverse effects for infants via breastfeeding. No drug-drug interactions have been identified [143].

Janus kinases (JAK) are non-receptor tyrosine kinase intracellular molecules that are involved in signaling pathways for cytokine receptors, such as IL-6 receptor, and growth factors. They are important in maintaining immune function [144]. Selective JAK inhibitors, such as baricitinib, inhibit the production of cytokines elevated in SARS-CoV-2, such as IL-2, IL-6, IL-10, and IFN-ɣ [145]. This drug has been shown to improve lymphocyte count and inhibit the entry of SARS-CoV-2 into cells [145]. A double blind randomized controlled study in 2021, showed that baricitinib combined with remdesivir was more effective in improving recovery times in SARS-CoV-2 infected patients who were receiving oxygenation [145]. The Food and Drug administration (FDA) has authorized the use of baricitinib in SARS-CoV-2 patients and pediatric patients >2 years old who are on supplemental oxygen. However, adverse effects of this medication should be noted. Most serious adverse effects are found in Rheumatoid arthritis patients who are on concomitant immunosuppressants. Adverse effects include cardiovascular disease, thrombosis, and malignancies such as lymphoma [146]. Additionally, adverse effects in SARS-CoV-2 patients include thrombocytosis, neutropenia, deep vein thrombosis, and pulmonary embolism. Baricitinib use is contraindicated with upadacitinib, as immunosuppressive risks are enhanced. Its use is not recommended with other potent immunosuppressive agents, JAK inhibitors, or disease modifying anti-rheumatic drugs (DMARD) [147].

Sotrovimab (VIR-7831) is a human neutralizing monoclonal antibody targeting the receptor binding domain (RBD) of the spike protein of SARS-CoV-2. The variable region of Sotrovimab binds to a highly conserved epitope of the SARS-CoV-2 spike protein in a region outside the highly mutagenic ACE2 receptor binding motif (Kd = 0.21 nM) and does not compete with human ACE2 receptor binding [148]. The highly conserved epitope targeted by Sotrovimab has been found on 99.8% of SARS-CoV-2 viruses which allows its use against mutagenic variants, including Omicron [149]. In May 2021, Sotrovimab received an Emergency Use Authorization in the United States for the treatment of mild to moderate SARS-CoV-2 in higher risk patients to prevent progression to severe symptoms [148]. Sotrovimab dosing is recommended within days of experienced symptoms. It inhibits viral fusion which ultimately reduces viral load and subsequently progression to severe symptoms [150]. The efficacy of Sotrovimab was studied by COMET-ICE investigators in a randomized, double-blind, placebo-controlled trial study. The enrolled symptomatic patients tested positive for SARS-CoV-2 and were at a higher susceptibility to progression of severe symptoms. A total of 291 patients in the treatment group received Sotrovimab 500 mg IV given once while 292 patients served as a control. The placebo group demonstrated 21 patients hospitalized for SARS-CoV-2 and one mortality. The Sotrovimab group only experienced three hospitalizations, with no cases of mortality, a relative risk reduction of 85%, and a statistically significant difference (*p* value: 0.002) [151]. Additionally, there were no significant differences in adverse effects between the two groups. Zheng et al. (2022), conducted a cohort study to compare the effectiveness of sotrovimab to molnupiravir in preventing severe SARS-CoV-2 outcomes in high-risk SARS-CoV-2 patients [152]. The study included 3288 patients receiving sotrovimab and 2663 receiving molnupiravir. While both groups showed a reduction in SARS-CoV-2 hospitalizations and deaths, patients with sotrovimab treatment were at a lower risk of severe SARS-CoV-2 symptoms compared to molnupiravir. There were 53 hospitalizations in the molnupiravir group and 31 in the sotrovimab group.

## 3. Conclusions

The role of GSH as an antiviral molecule has been explored in studies regarding SARS-CoV-2 and Influenza. A study performed by De Flora et al. showed that patients who received doses of N-acetylcysteine, a precursor of GSH, for six months had a significant decrease in clinically apparent disease [153]. A smaller study demonstrated that SARS-CoV-2 patients with a lower baseline GSH level had more severe symptoms while SARS-CoV-2 patients with a higher baseline GSH level had milder symptoms [154]. GSH is well known for its anti-inflammatory functions while SARS-CoV-2 is known to exacerbate inflammatory processes within the body. The potential use and functions of GSH to enhance the immune response against SARS-CoV-2 represent a topic that warrants further discussion.

One of the major protective effects of GSH on inflammation is driven by the modulation and balance of ACE/ACE2 activity in cells infected by SARS-CoV-2. The upregulation of ACE activity and concurrent downregulation of ACE2 activity leads to the production of ROS within the cell [71]. The production of ROS and subsequent change in cellular redox status increases activity of NF-kB [155]. In a study using SARS infected mice, it was shown that drugs which inhibit NF-kB signaling reduce inflammation and increase survival in mice after infection with SARS-CoV [156]. GSH has inhibitory effects on the activity of ACE and has the ability to decrease the production of ROS via inhibition of ACE, leading to decreased NF-kB signaling, providing an avenue for decreased inflammation in SARS-CoV infected cells [157].

## Figures and Tables

**Figure 1 ijms-23-09297-f001:**
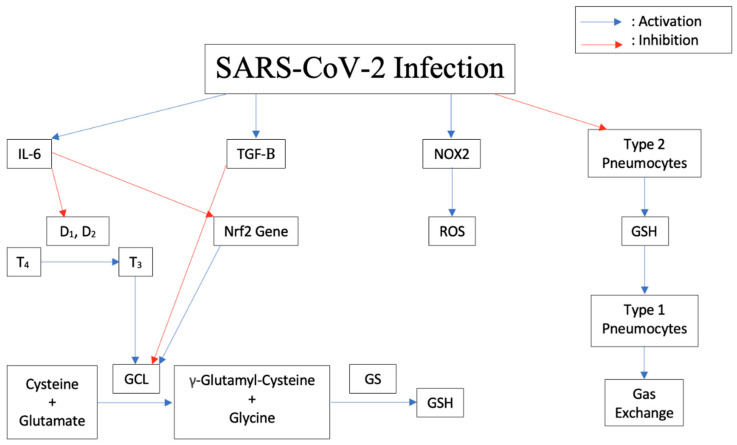
Downstream effects of SARS-CoV-2 Infection on cytokine release, thyroid conversion, ROS production, GSH inhibition, and pneumocyte damage in humans.

**Figure 2 ijms-23-09297-f002:**
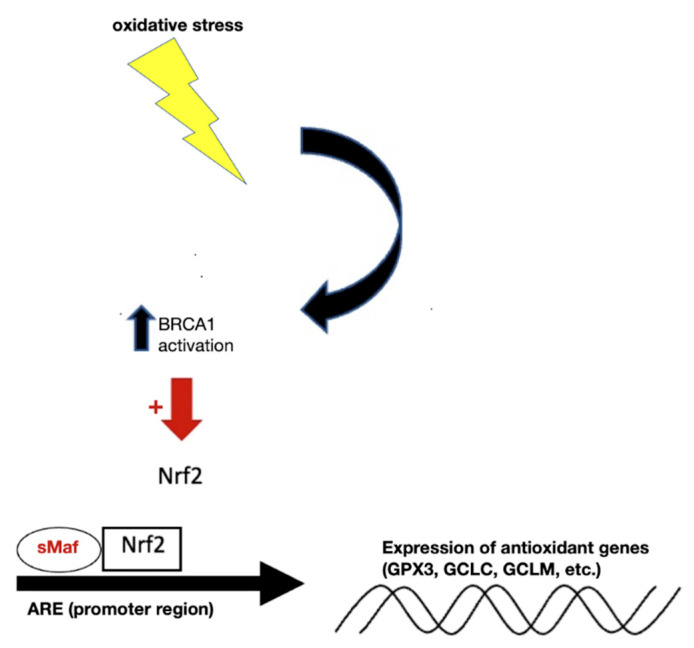
Under normal conditions, BRCA1 located in the nucleus facilitates the expression of Nrf2, leading to upregulation of the ARE to produce antioxidant genes important for GSH synthesis. Abbreviations: BRCA1, breast Cancer gene 1; Nrf2, nuclear factor erythroid 2–related factor 2; ARE, antioxidant responsive element; GPX3, glutathione peroxidase 3; GCLC, glutamate-cysteine ligase catalytic subunit; GCLM, glutamate-cysteine ligase regulatory subunit.

**Figure 3 ijms-23-09297-f003:**
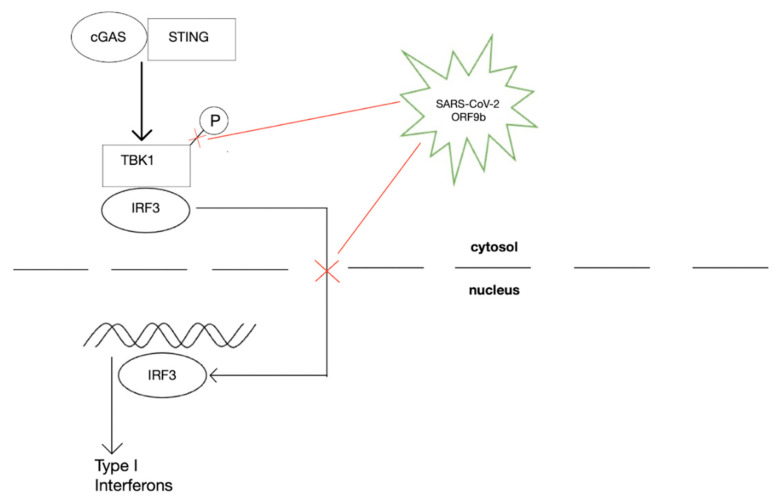
cGAS-STING pathway inhibition by SARS-CoV-2 ORF9b which consequently reduces normal production of type I interferons. Abbreviations: cGAS, cyclic GMP-AMP synthase; STING, stimulator of interferon genes; TBK1, TANK-binding Kinase 1; IRF3, Interferon Regulatory Factor 3.

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
