# Peer review of "Recent Developments in the Understanding of Immunity, Pathogenesis and Management of COVID-19"

_ijms, 2022, doi:10.3390/ijms23169297_

Round 1
Reviewer 1 Report
Competent, well written review.
Remarks:
The author could consider inserting a table documenting the decrease in GSH content in various viral diseases including the cells/tissues where the GSH level was quantified. It is a suggestion but not a requirement; the review is well balanced in the submitted form.
Fig. 1 Please explain D1, D2
Line 232: “was severely diminished”, wouldn’t “is severely diminished” be better?
Line 240: which isoenzyme of superoxide dismutase?
Lines 276-279: NADPH is required rather for GSH regeneration (from GSSG, by glutathione reductase) than GSH generation
Author Response
Cover letter: Recent Developments in the understanding of immunity, pathogenesis and management of COVID-19.
REVIEWER#1
Thank you for reviewing our paper and suggesting the edits stated below, we have looked at all the suggestions and made the necessary revisions as requested by reviewer 1 and 2.
The following comments/remarks:
The author could consider inserting a table documenting the decrease in GSH content in various viral diseases including the cells/tissues where the GSH level was quantified. It is a suggestion but not a requirement; the review is well balanced in the submitted form.
- Thank you for the suggestion it is a great point but we have decided to not include a table at the time.
Fig. 1 Please explain D1, D2
- Added parenthesis in the text to explain what they are in the figure.
Line 232: “was severely diminished”, wouldn’t “is severely diminished” be better?
- Changed “was” to “is”
Line 240: which isoenzyme of superoxide dismutase?
- Changed to Superoxide dismutase isoform 1 (SOD1) to be more specific
Lines 276-279: NADPH is required rather for GSH regeneration (from GSSG, by glutathione reductase) than GSH generation
- Changed to “plays a role in the regeneration of GSH”

Reviewer 2 Report
The authors presented a comprehensive review on the COVID-19. They touch the problem from various sides and consider its different aspects. The only thing I would add to this paper, is some more information about the latest mutations of the omicron variant and also on the changing symptoms/mortality with the subsequent dominating variants of the virus. I would also like to congratulate authors on their great work.
Author Response
Cover letter: Recent Developments in the understanding of immunity, pathogenesis and management of COVID-19.
Thank you for reviewing our paper and suggesting the edits stated below, we have looked at all the suggestions and made the necessary revisions as requested by reviewer 1 and 2.
REVIEWER#2
The authors presented a comprehensive review on the COVID-19. They touch the problem from various sides and consider its different aspects. The only thing I would add to this paper, is some more information about the latest mutations of the omicron variant and also on the changing symptoms/mortality with the subsequent dominating variants of the virus. I would also like to congratulate authors on their great work.
- Added to the paragraph in section 2.2 with further information on the omicron variant.
